# Porphyrin Functionalized Carbon Quantum Dots for Enhanced Electrochemiluminescence and Sensitive Detection of Cu^2+^

**DOI:** 10.3390/molecules28031459

**Published:** 2023-02-02

**Authors:** Xinying Zhang, Xialing Hou, Decheng Lu, Yingying Chen, Lingyan Feng

**Affiliations:** Materials Genome Institute and Shanghai Engineering Research Center of Organ Repair, Shanghai University, Shanghai 200444, China

**Keywords:** carbon quantum dots, porphyrin, electrochemiluminescence, Cu^2+^ detection

## Abstract

Porphyrin (TMPyP) functionalized carbon quantum dots (CQDs-TMPyP), a novel and efficient carbon nanocomposite material, were developed as a novel luminescent material, which could be very useful for the sensitive detection of copper ions in the Cu^2+^ quenching luminescence of functionalized carbon quantum dots. Therefore, we constructed a sensitive “signal off” ECL biosensor for the detection of Cu^2+^. This sensor can sensitively respond to copper ions in the range of 10 nM to 10 μM, and the detection limit is 2.78 nM. At the same time, it has good selectivity and stability and a benign response in complex systems. With excellent properties, this proposed ECL biosensor provides an efficient and ultrasensitive method for Cu^2+^ detection.

## 1. Introduction

Carbon-based nanomaterials have become ideal materials for the construction of biosensors due to their advantages of low toxicity [1], good biocompatibility [2], and easy functionalization and modification [3]. Carbon quantum dots, as novel carbon-based nanomaterials, not only have the advantage of carbon nanomaterials, but have some unique qualities such as florescence and ECL properties. Because of its easy synthesis, carbon quantum dots have become a good functional nanomaterial for biosensors. Electrochemiluminescence (ECL) is a type of analytical method that combines the electrochemical and florescent methods. ECL biosensors have some advantages such as lower background signal. In this way, many nanomaterials such as gold nanoclusters and sliver nanoclusters present ECL properties. Carbon-based nanomaterials have attracted a lot of attention for ECL functional materials. In 2009, Zheng et al. found that carbon quantum dots had an ECL phenomenon through electrolytic graphite rods, opening a new chapter of carbon-based nanomaterials as ECL luminescent species [4]. However, the ECL luminescence of carbon-based nanomaterials is insufficient, so the development of new carbon-based nanomaterials with high ECL performance has become a common concern. Carbon based quantum dots (CQDs) have attracted wide attention as a non-toxic, environmentally friendly, and cheap material [5]. In the past few years, most of the research on carbon quantum dots (CQDs) has focused on the synthesis and properties of CQDs [6]. As a new type of probe, CQDs do not show obvious advantages in the sensitivity and selectivity of ECL sensors, mainly due to the weak ECL emission intensity of CQDs, which leads to low detection sensitivity [7], or the CQDs are not functionalized, resulting in poor selectivity [8]. Therefore, it is very important to develop a special functionalized CQD ECL probe with high luminescence efficiency for the highly sensitive analysis and detection of ECL.

Porphyrin is a kind of macromolecular heterocyclic compound which is composed of four pyrrole subunits α- Carbon atoms are formed through the interconnection of methylene bridge [9]. Porphyrins have unique and highly stable aromatic heterocyclic structures, and their electrochemical and photochemical properties have attracted attention for decades, where they are often used as photosensitizers and optical probes [10]. Through chemical modification, porphyrins have flexible and adjustable molecular structures, which can enable them to form new nanostructures [11]. However, due to the low solubility and instability of the intermediate and the narrow potential window, it is difficult to achieve efficient and stable ECL emission in an aqueous solution. To improve the ECL performance of porphyrins, many methods have been designed. Because porphyrins are 18π aromatic azacycienes, they can assemble with the help of π-π stacking and electrostatic action. The aggregation of porphyrins has two styles. H aggregation is parallel to each other. When they move vertically along its surface, porphyrins overlap with adjacent molecules. On the other hand, J aggregation is dislocation parallel aggregation. The porphyrins will overlap with adjacent porphyrins only when they move vertically along their plane and then moving parallel. Porphyrin aggregation processed a unique aggregation introduced emission. The TCPP J aggregate performed good ECL intensity [12]. A novel designed Zn-coordination covalent porphyrin was assembled with benzylamine to enhance the ECL intensity [13]. ZnTCPP could assemble into MOF with the help of methylimidazole. In this way, this MOF exhibited excellent ECL performance [14]. Porphyrins also aggregate when combined with other materials; graphene is combined with porphyrin to form a new porphyrin nanosphere–graphene oxide composite that has great ECL properties [15]. Carbon nanotubes are a good platform for assembling porphyrins to process excellent ECL [16]. Chiral carbon dots introduced the chiral self-assembly of porphyrins [17]. Furthermore, sometimes fluorescence resonance energy transfer exited in the complex of porphyrins and carbon dots [18]. Therefore, the carbon dots could increase the fluorescence intensity of porphyrins.

Excessive heavy metal ions such as gold ions, sliver ions, and copper ions cause great harm to the human body. Heavy metal ions are generally difficult to degrade and can enter the organism through the cyclic enrichment of the ecosystem, which seriously threatens people’s health and the environment [19]. Copper ions are one of the basic elements of both humans and plants, and abnormal concentrations of Cu^2+^ will lead to neurodegenerative diseases such as Alzheimer’s disease, Parkinson’s disease, and Wilson’s disease [20]. Therefore, high selectivity and sensitivity to trace Cu^2+^ in biological samples is of great significance. While the detection of Cu^2+^ still has some problems, finding new receptors that have a high selectivity for Cu^2+^ is the common goal. Because the concentration of Cu^2+^ is very low in the human body, it is very important to develop highly sensitive biosensors for Cu^2+^ detection, while the complex environment will disturb the accuracy of the detection results.

Based on the above research results and the current situation, in this section, we prepared negatively charged nitrogen carbon quantum dots by the hydrothermal method, and combined positively charged 5, 10, 15, 20-tetrakis (1-methyl-4-pyndno) porphyrin tetra (p-toluenesulfonate) (TMPyP) with the electrostatic interaction between them to obtain the porphyrin molecular functionalized carbon quantum dots (CQDs-TMPyPs), as shown in Figure 1A. CQDs-TMPyPs showed strong ECL emission and excellent stability, and an ECL sensor was constructed based on the quenching of CQDs-TMPyPs by heavy metal ions of Cu^2+^. As shown in Figure 1B, CQDs-TMPyPs were modified on the surface of the glass carbon electrode and showed good ECL luminescence in the system with sodium persulfate as the co-reactant. When Cu^2+^ was added, the CQDs-TMPyPs combined with amino acids on the surface of CQDs to form copper amine, leading to selective and strong ECL quenching [6]. Therefore, they are sensitive to the concentration of Cu^2+^.

## 2. Results and Discussion

### 2.1. Characterization of Nanomaterials

Porphyrins can easily aggregate together due to their chemical structure, which is beneficial for π-π stacking and electrostatic interaction. Because the carboxyl terminal of CQDs in a neutral solution environment (pH = 7.4) is negatively charged. TMPyP is a tetrapationic porphyrin, so TMPyP is positively charged in a neutral solution. Therefore, it can be easily combined with CQDs to form the CQD-TMPyP conjugate through electrostatic interaction [21]. In order to prove the aggregation between TMPyP and CQDs, we first characterized CQD–TMPyP by analytical techniques. CQDs, TMPyP, and their composites were characterized by FTIR (Figure 1A). Infrared spectra showed the presence of -NH_2_, -SH, -COOH, and -OH groups. The peaks of 1570, 1507, and 1453 cm^−1^ were attributed to the stretching vibrations of the carbon skeleton of PAHs. The peak of 3325 cm^−1^, which appeared in the spectra of porphyrins, represents the stretching vibration absorption peak of the N-H of porphyrin. The bending vibration absorption peak of N-H appeared at 965 cm^−1^, which implies that there was no metal ion combination. The peaks at 3050 and 3400 cm^−1^ were due to the C-H/N-H/O-H stretching vibrations. The peak at 1709 cm^−1^ was derived from the C=O/COO- stretching, while the wide band at 1200 and 1310 cm^−1^ corresponded to the asymmetric stretching vibration of C-C/C-N/C-O, indicating that the surface of the CQDs contained a large number of -COOH and -OH groups. The IR spectra of CQDs presented a peak at 1450 cm^−1^, which was caused by C-S bonds. This suggests that CQDs contain a sulfur element. The same peak appeared in the IR spectra of CQD-TMPyP, indicating that CQDs were included in the CQD-TMPyP. The 3400, 3050, and 1709 cm^−1^ peaks can be seen in the IR spectra of CQD-TMPyP. Therefore, the CQD-TMPyP contained CQDs. The peaks at 1700, 1200, and 1310 cm^−1^ also appeared in the IR spectra of CQD-TMPyP, which indicates that there was no chemical change in the surface of the CQDs in the nanocomposite CQD-TMPyP. The aggregation process of the CQD-TMPyP did not have a chemical reaction, and electrostatic interaction was the main driving force of assembly between CQDs and TMPyP. Figure 1B shows the characterization of the ultraviolet spectrum. In this figure, the shoulder peak of the composite material at ~290–300 nm (C=O n-π* electron transfer) can be seen, and the characteristic porphyrin peak appeared to red shift at 421 nm (π-π* interaction). The UV absorbance peak of the CQDs was 260 nm. There was no peak at 421 nm, which appeared in the UV spectra of TMPyP, while CQD-TMPyP had two absorbance peaks at 260 nm and 421 nm, indicating a good assembly between CQD and TMPyP.

Transmission electron microscope (TEM) images were used to characterize the morphology of the CQDs (Figure 2A). The size of the prepared CQDs was relatively uniform, with an average diameter of 2 nm. We used AFM to prove the aggregation of CQDs-TMPyP. In Figure 2B, the AFM height difference showed that the size of the CQDs was about 6 nm, while the composite CQDs-TMPyP (Figure 2C) was about 16 nm. This indicates that the CQDs-TMPyP composite material was formed by the combination of the two compounds through electrostatic interaction. The positive porphyrins reduced the electrostatic repulsion between the carbon quantum dots. Then, the complex between the CQDs and TMPyP aggregated again to form larger composites.

The fluorescence response energy transfer also appeared in the CQDs-TMPyP. In Appendix A, the CQDs showed a fluorescence emission at 450 nm with an excitation peak at 370 nm. The quantum yield of the CQDs was 0.53%. The fluorescence lifetime of the CQDs was also measured, which was 2.1 ns. After being combined with TMPyP, the quantum yield of CQDs-TMPyP was calculated as 1.04%. The fluorescence lifetime of CQDs-TMPyP was 4.7 ns. TMPyP showed fluorescence at 700 nm, as shown in Appendix A. The excitation peak of TMPyP was 480 nm. The fluorescence of CQDs-TMPyP with an excitation peak at 370 nm is shown in Appendix A. The CQDs-TMPyP presented two emission peaks at 450 nm and 700 nm. Thus, CQD-TMPyP maintains the properties of CQDs and TMPyP. As shown in Appendix A, following the addition of TMPyP into CQDs, the emission peak at 450 nm, which is the emission of CQDs, decreased, but the emission of TMPyP at 700 nm increased, which is the result of the fluorescence response energy transfer between the CQDs and TMPyP. This indicates the good assembly between CQDs and TMPyP. The influence of different excitation peaks for the emission peaks of the CQDs and CQD-TMPyP composite can be observed in Appendix A. The emission peak of CQDs showed no obvious change during the conversion of the excitation peak from 300 nm to 380 nm in Appendix A, indicting the uniform size of the CQDs. The fluorescence intensity of the CQDs increased with the redshift of the excitation peak. The optimum excitation wavelength was 370 nm. The CQDs-TMPyP showed two clear emission peaks at 450 nm and 700 nm at different excitation peaks. This implies that the CQDs assembled with the porphyrins successfully. Following the excitation peaks from 300 nm to 400 nm, CQD-TMPyP showed a stable emission performance in Appendix A. The change in the excitation wavelengths only caused an increase in the emission intensity.

### 2.2. ECL Mechanism

The ECL properties of different luminescent species CQDs and TMPyP and the composite nanomaterial CQDs-TMPyP were studied. Figure 3 shows that the ECL emission generated during the cyclic potential scanning from 0 to −2 v in 0.1 M Na_2_S_2_O_8_ buffer. The CQDs and TMPyP showed a weaker ECL emission. In contrast, the self-assembled composite CQDs-TMPyP showed about 5 to 10 times enhanced ECL compared with the CQDs and TMPyP. This was due to the fact that hydrophilic carbon dots (CQDs) can effectively prevent the dissolution of the TMPyP porphyrin in aqueous solutions and accelerate electron transfer [16]. The fluorescence resonance energy transfer could be another reason for the increase in the ECL phenomenon. Therefore, CQDs-TMPyP/GCE achieved an enhanced ECL emission with Na_2_S_2_O_8_ as the co-reactant.

According to the cathode ECL reports of CQDs and TMPyP [10,22] and the experimental results, the possible ECL mechanism is described by the following equation. The electrochemical reaction between CQDs and TMPyP occurs on the surface of the glassy carbon electrode, and electrons are injected into the working electrode to generate CQDs (CQDs^•−^) and TMPyP (TMPyP^•−^). At the same time, S_2_O_8_^2−^ is electrically reduced to SO_4_^2−^ and SO^•−^. Subsequently, the highly oxidized SO^•−^ reacts with the reduced species CQDs (CQDs^•−^) and TMPyP (TMPyP^•−^) to generate the excited states CQDs* and TMPyP*. Finally, CQDs* and TMPyP* return to the ground state CQDs and TMPyP produces the ECL emission.

### 2.3. Optimization of Experimental Conditions

Experimental conditions have a significant impact on the analysis and detection performance of this sensor. Therefore, the potential window and pH value imposed during the experiment were optimized to explore the influence of different experimental conditions on the biosensor ECL signal, respectively. As shown in Figure 4, the ECL signal gradually increased with the increase in the negative voltage applied at both ends of the electrode when the potential window was within 0 to −2 V. When the potential was lower, the ECL signal decreased. When the potential was higher, the ECL signal at the potential of 0~2.2 V was higher than that at the potential of 0~2.1 V. However, considering the influence of the electrolysis of water, the potential application in an ECL system is not very high. The potential window of 0 to −2 V was the best potential window of this ECL system, showing the maximum ECL intensity. The CQD assembly with TMPyP had the help of electrostatic interaction. The pH of the reaction solution is an important factor for the ECL intensity of CQD-TMPyP. As can be seen in Figure 4B, the ECL signal gradually increased as the pH value increased from 5.0 to 7.0. However, it showed a downward trend in the range of 7.0~9.0 because the neutral conditions were more conducive to the electrostatic induction between TMPyP and CQDs, so the ECL emission of the system showed the best. Therefore, the optimal experimental conditions of the potential window from 0 to −2 V and pH 7.4 were selected to test the sensor performance. The potential scanning rates are also discussed in Appendix A. The ECL intensity of the CQD-TMPyP increased with the increase in the scanning rates. We chose 0.1 Vs^−1^ as the testing scan rate.

### 2.4. Performance of the ECL Biosensor

Under the optimized experimental conditions, as shown in Figure 5A, the ECL signal of CQDs-TMPyP quenching could be obviously observed after adding different concentrations of Cu^2+^ to it. With the increase in the Cu^2+^ content, the ECL signal gradually decreased. However, the ECL signal remained stable after adding different concentrations of Cu^2+^. This quenching phenomenon of CQDs-TMPyP was due to the reaction of the CQDs-TMPyP surface functional groups (C-OH, -COOH, C-O-C, -NH_2_) with metal ions. The surface of CQDs contained abundant amino groups, which can combine Cu^2+^ ions with -NH_2_ to form copper amines. The absorption of the excited and/or emitted light by the absorbent might reduce the intensity of the luminescence group, thus quenching the filtering effect of the emitted light. As shown in Figure 5B, there was a good linear relationship between the signal change value of ECL and the logarithm of Cu^2+^ concentration from 10 nM to 10 μM, and the detection limit was calculated to be 2.78 nM (detection limit was the concentration of Cu^2+^ whose ΔI was three times that of the ECL intensity without the target Cu^2+^, S/N = 3). The linear relationship can be expressed as ΔI = 37.22 logc − 30.33, and the correlation coefficient (R^2^) was 0.99. (ΔI = (I_0_ − I)/I_0_, where I_0_: ECL strength without target Cu^2+^, I: ECL strength after adding different concentration of Cu^2+^). The results showed that the proposed ECL sensors can realize the sensitive detection of Cu^2+^. In addition, compared with various biosensors with other nanomaterials for the detection of Cu^2+^, as shown in Table 1, the ECL sensor constructed by us had a much lower detection limit and a wider detection range for Cu^2+^.

### 2.5. Stability and Selectivity

The selectivity, the difference between the target signal and jamming signal of biosensors, is another important index to evaluate the sensor performance. In order to evaluate the selectivity of the constructed biosensor, we carried out ECL control experiments with Cd^2+^, Zn^2+^, Al^3+^, Co^2+^, Fe^2+^, Ba^2+^, Ni^+^, K^+^, and Mg^2+^ under the same experimental conditions. As shown in Figure 5C, Cd^2+^, Zn^2+^, Al^3+^, Co^2+^, Fe^2+^, Ba^2+^, Ni^+^, K^+^, and Mg^2+^ showed extremely low signal strength, even at a 10 μM concentration. The signal change produced by Cu^2+^ was 4 to 5 times higher than the others, which was produced by adding other metal ions in this system. There is no doubt that this biosensor has good selectivity for Cu^2+^. In the presence of Cu^2+^, the ECL intensity increased significantly and could be distinguished from other interferences. The results showed that the biosensor had good specificity for Cu^2+^. Stability is also an important factor for biosensors, so we evaluated the stability of the ECL biosensor. The common method for ECL biosensors to detect their stability is to measure the ECL intensity in the circle. The ECL response of this biosensor at different times is shown in Figure 5D. The ECL intensities measured before 400 s were nearly the same. After 400 s, the ECL response of CQDs-TMPyP slightly decreased. Therefore, CQDs-TMPyP appears to have a stable ECL response.

### 2.6. Real Sample Analysis

The performance of biosensors in a complex environment is a very important factor for the follow-up clinical application of biosensors. In order to test the performance of this designed sensor in disease testing, different concentrations of Cu^2+^ were added to bovine serum samples diluted by 10 times to simulate the serum environment of real samples. The standard recovery experiment was carried out. Three parallel experiments were measured in this section. The ECL intensities of this biosensor were recorded in the presence of bovine serum samples with different concentrations of Cu^2+^. The recovery was the ratio of concentration of Cu^2+^ calculated and the concentration added into the bovine serum. As shown in Table 2, the RSD (relative standard deviation) was less than 8.20%, and the recovery rate ranged from 98% to 122.7%, which proved that the biosensor still had a good response in complex environments.

## 3. Materials and Methods

### 3.1. Reagents and Materials

Cysteine (L-Cys) and sodium hydroxide (NaOH) were purchased from Sigma-Aldrich Trading Co. Ltd. (Shanghai, China). Sodium persulfate (Na_2_S_2_O_8_, ≥98%), porphyrin (TMPyP) was purchased from Aladdin Reagent Co. Ltd. (Shanghai China). Disodium hydrogen phosphate (Na_2_HPO_4_), sodium dihydrogen phosphate (NaH_2_PO_4_), and sodium chloride (NaCl) were purchased from Sigma-Aldrich (Shanghai, China). A solution of 0.1 M PBS (pH 7.4) containing 0.1 M Na_2_S_2_O_8_ and 0.1 M NaCl was used as the electrolyte in the ECL analysis. Normal fetal bovine serum was purchased from Sangon Biotech (Shanghai, China). All reagents were analytical pure grade and used without further purification. All aqueous solutions were prepared using ultrapure water (18.2 MΩ, Milli-Q, Millipore, Burlington, MA, USA).

### 3.2. Apparatus

ECL measurements were performed with a BPCL-Q-GP21-TGC Ultra-Weak luminescence analyzer with aa conventional three-electrode cell consisting of a platinum wire as the auxiliary, Ag/AgCl (3 M KCl) as the reference electrode, and the modified GCE as the working electrode. Unless otherwise noted, the PMT was 1100 V and the potential scan was from 0 V to −2 V at a scan rate of 0.1 V·s^−1^. Electrochemical measurements were investigated on an Autolab potentiostat/galvanostat instrument (Metrohm China Ltd., Herisau, Switzerland) (Shanghai, China). UV absorbance measurements were carried out on a JASCO V-750 UV-Vis spectrophotometer (Toshi, Japan) and FTIR characterization was measured on a Thermo Fisher Nicolet Is50 FTIR spectrometer (Waltham, MA, USA). The transmission electron microscopy (TEM) images were finished using a JEM-200CX transmission electron microscope (Osaka, Japan) and AFM was performed by a Micronano D5A Scanning Probe Microscope (Shanghai Zoolon Micro and Nano Equipment Co., Ltd. Shanghai, China).

### 3.3. Synthesis of the CQD-TMPyP Nanocomposite

Nitrogen and sulfur doped carbon quantum dots with negative charge were prepared by the hydrothermal method in one step as in [31]: 0.5 g of cysteine (L-Cys) and 0.6 g of sodium hydroxide (NaOH) were dissolved in 15 mL of water. In order to fully dissolve the sample, this mixture was mixed ultrasonically for 30 min. Then, the obtained solution was transferred to a polytetrafluoroethylene hydrothermal kettle and heated at 120 °C for 16 h. After that, the cellulose dialysis membrane, whose molecular retention was 1 KDa, was used for purifying the brown solution for three days. Carbon quantum dots were obtained after the removal of impurities.

Under the condition of avoiding light, the porphyrin solid was dissolved in the aqueous solution, and its ultraviolet absorption was tested. The width of the dish L was 1.0 cm. According to the formula “A = ε · C · L”, the concentration of the configured TMPyP solution C was 2.1301 × 10^−4^ M. Among them, for A solution at 422 nm absorbance, the light absorption coefficient of the solution was “ε = ${2.66\;\times \;10}^{-5}\;M^{-1}\cdot {\mathrm{cm}}^{-1}$. At room temperature, 10 μM TMPyP porphyrin solution was added to 18.75 μg/mL carbon quantum dot solution and mixed for 2 min, so that the two compounds were fully combined by electrostatic interaction to form a CQD-TMPyP composite ECL probe [32].

### 3.4. Electrode Cleaning and Preparation

Prior to use, the glassy carbon electrodes (GCE, ø = 3 mm, CHI) were polished with 0.3, and 0.05 μm alumina powder for one minute and then sonicated for 3 min. After 10 min of sonication, glassy carbon electrodes were put into a potassium ferricyanide solution to measure the cyclic voltammetry in order to evaluate the performance of the electrodes. We selected the electrodes with the difference of the redox peak potential within 80 mV as electrodes for further modification. Then, 5 μL of the CQD-TMPyP nanocomposite was dropped on the pre-treated GCE and dried at 37 °C overnight.

### 3.5. ECL Measurements

ECL measurements were performed in detection buffer (100 mM PBS, 100 mM NaCl pH = 7.4) containing 0.1 M Na_2_S_2_O_8_ at a scan rate of 0.1 V·s^−1^ with a potential from 0 to −2 V under a PMT voltage at 1100 V. The modified electrodes were incubated with different concentrations of Cu^2+^. Then, the ECL and CVs curves were recorded simultaneously. All the experiments had three or more parallel experiments.

## 4. Conclusions

In summary, we prepared a novel nanocomposite CQDs-TMPyP through electrostatic interaction by a simple mixing method without any chemical modification. This nanocomposite successfully overcame the shortcomings of the application of CQDs in ECL. Without further chemical modification, CQDs-TMPyP showed an excellent ECL response and selective response for copper ions. It provided a new method to overcome the shortage of CQDs for the construction of ECL biosensors such as low ECL intensity and poor selectivity. After that, an ultrasensitive electrochemiluminescence biosensor for Cu^2+^ was successfully developed using CQDs-TMPyP as an ECL functional nanomaterial. Based on its excellent ECL response, this biosensor made of CQDs-TMPyP realized the sensitive detection of Cu^2+^ by the quenching action of Cu^2+^ ions for CQDs-TMPyP. Compared with the existing Cu^2+^ ion sensors, its detection range is wider as the detection limit can reach the nmol⸱L^−1^ level. With this biosensor design, we achieved the aim for the highly selective and ultra-sensitive detection of Cu^2+^ with a detection limit as low as 2.78 nM. In addition, this biosensor has a strong ability to analyze real samples and has a significant signal response in the serum samples, which is expected to be a new method for the early diagnosis of disease and environmental monitoring.

## Data Availability

The data are available upon reasonable request.

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
