# Peer review of "Porphyrin Functionalized Carbon Quantum Dots for Enhanced Electrochemiluminescence and Sensitive Detection of Cu2+"

_molecules, 2023, doi:10.3390/molecules28031459_

Round 1
Reviewer 1 Report
This manuscript deals with the hydrothermal synthesis of nitrogen and sulfur dopped carbon quantum dots negatively charged from cysteine and sodium hydroxide. These CQDs were funtionalized with the positively charged TMPyP porphyrin by electrostatic interactions. The supramolecular assembly TMPyP-CQDs was used to construct an ECL sensor based on the ECL signal quenching of TMPyP-CQDs by Cu2+ ions with interesting results and low detection limits. However, there are few elements that are vague and/or lack sufficient analysis and justification. Specifically, the following issues need to be addressed in a revised manuscript:
In my opinion, the introduction needs to be reworked because some paragraphs are quite difficult to understand.
Page 2, the sentence from line 56 to line 59 it is not clear and in my opinion should be rewrite.
Page 2, line 76: change spectrums for techniques.
Page 2, Scheme 1: It would be very useful for the global comprehension of the present study, to add the structure of the TMPyP porphyrin and a schematic representation of its interaction with the CQDs.
Page 3, figure 2: the scale bars of B and C images are missing.
CQDs are nanomaterials with good fluorescence properties. However, in the present work no emission data is given. I suggest to perform an emission profile of aqueous dispersion of CQDs under different excitation wavelengths at pH 7.4, and the same experiment with the TMPyP-CQDs assemble in order to complete the characterization of the pristine and functionalized materials.
I think that the authors should calculate the quantum yield of these new N-CDs, to have a data that could be compared with other N-CDs in the literature.’
Do you have an idea of the amount of TMPyP porphyrin anchored on the CQDs, approximately?
Do you observe aggregation of the TMPyP-CQDs assemblies? In TEM you observe a very large increase from the raw CQDs material (6 nm) to the TMPyP-CQDs assemblies (16 nm). Do you have an explanation for that?
Page 5, figure 5: In the caption “from 10 nM to 10 µM” µ is missing.
Reviewer 2 Report
The electrical characteristics of porphyrin and nitrogen carbon quantum dots was used to prepare the porphyrin molecular functionalized carbon quantum dots (TMPyP-CQDs). TMPyP-CQDs exhibited good ECL luminescence in the system with sodium persulfate as co-reactant and achieved selective quenching through the formation of copper amines. The sensitivity and low detection limits of the sensor are of interest, the following modifications should be made to meet the journal's requirements for publication:
1. I recommend using abbreviations of words with caution. Abbreviations should be marked in parentheses when the word first appears, while keywords should be used in full unless the abbreviation of the word is commonly accepted in the industry.
2. Please check the clarity and size of all figures to ensure they are suitable for the reader.
3. The title of the figure should be straightforward and easy to understand. Please revise the title of Figure 3 and check the whole text, vs. potential whether it is a spelling mistake or a special usage.
4. For the description in lines 124 to 131, the meaning of M should be specified. The content of this section had not been verified as necessary except for the references, please reconsider whether this section should occupy so much space in the 2. Results and discussion section or appear in the text, and whether it is repeated with the description in the previous paragraph.
5. Has the author considered the potential window beyond 0~-2.2V, because I noticed that 0~-2.2V is better than 0~-2.1V.
6. In Figure 5, the stability of the sensor should provide a suitable reference indicator with the corresponding value.
7. I strongly recommend that authors check the format of the manuscript for compliance with the journal's requirements and make any necessary corrections.
Round 2
Reviewer 1 Report
The authors have addressed all the questions and comments.